# Allelopathic Effects of Dominant Native Invaders on Forage Establishment: Implications for Alpine Meadow Restoration on the Qinghai-Xizang Plateau

**DOI:** 10.3390/plants14223506

**Published:** 2025-11-17

**Authors:** Xin Liu, Yaojun Ye, Zaihong Yang, Yazhou Zhang

**Affiliations:** 1School of Ecology and Environment, Southwest Forestry University, Kunming 650224, China; liuxinlm521@163.com; 2Yunnan Key Laboratory of Plant Diversity and Biogeography, Kunming Institute of Botany, Chinese Academy of Sciences, Kunming 650201, China; yeyaojun624@163.com

**Keywords:** Qinghai-Xizang Plateau, allelopathy, seed germination, phenolic acid, native plant invasion

## Abstract

The expansion of native invasive plants severely impacts alpine meadow ecosystems and regional development on the Qinghai-Xizang Plateau by reducing vegetation productivity and hindering livestock production. However, the rules underlying their effects on forage grass establishment and effective mitigation strategies remain poorly understood. Here, using three main allelochemicals—benzoic acid (BA), caffeic acid (CA), and p-hydroxybenzoic acid (HA)—from typical native invasive plants, we investigated concentration-dependent effects (0, 100, 300, and 500 mg/L) on the seed germination and seedling growth of four common forage species: *Festuca elata* Keng ex E. B. Alexeev (FE), *Lolium perenne* L. (LP), *Medicago sativa* L. (MS), and *Trifolium repens* L. (TR). Our findings revealed a concentration-dependent hormesis effect: low concentrations stimulated germination and growth, while inhibition intensified with increasing concentrations. Roots exhibited significantly higher sensitivity than stems (*p* < 0.01). The phytotoxic intensity of allelochemicals on forage grass growth follows the order BA > CA > HA. For germination (germination rate/potential), sensitivity orders were FE > LP > TR > MS and LP > FE > TR > MS, respectively. For seedling growth, toxicity orders were TR > MS > FE > LP (root length), TR > FE > MS > LP (root weight), TR > MS > FE > LP (stem length), and TR > FE > LP > MS (stem weight). In summary, different allelochemicals exerted significantly varied effects on the germination and growth of distinct forage grass species. Therefore, forage species selection should consider local allelochemical profiles, or alternatively, grass-legume mixtures could be employed to enhance biomass yield. Our findings provide valuable insights for developing effective grassland restoration strategies.

## 1. Introduction

Biological invasions pose significant threats to global ecosystems, resulting in substantial economic losses and degradation of ecosystem functioning [1,2]. While traditional definitions of biological invasion have primarily focused on non-native/alien plants crossing geopolitical boundaries (e.g., country), increasing evidence suggests that native invasive plants, driven by anthropogenic activities such as intensive overgrazing and land conversion, can exhibit aggressive expansion behaviors [3,4]. A plant’s invasive status should be defined by its ecological impact rather than origin, as even naturally occurring species may exhibit non-invasive traits in stable habitats yet become aggressively invasive in disturbed adjacent ecosystems—a phenomenon termed “native invasion” [5,6]. Despite causing ecological and economic damage similar to that of non-native invasive species, native invasive species remain largely overlooked within conventional invasion biology frameworks and subsequent management policies [5,7]. This conceptual gap hinders effective ecosystem restoration and conservation strategies tailored to this specific threat. Consequently, broadening the scope of biological invasion theory to encompass native plant invasions is imperative [8]. Investigating the ecological impacts of these native invasions, and developing targeted mitigation strategies holds significant scientific and practical importance [9]. This phenomenon is of particular concern on the Qinghai-Xizang Plateau (QXP), a region serving as a critical ecological barrier with extensive alpine meadow ecosystems [10]. Local pastoral communities are heavily dependent on these meadows for livestock grazing [11]. In recent years, however, the proliferation of native invasive plants has accelerated, correlating with severe degradation of meadow ecosystems. And the area and nutritional value of edible feed species have decreased significantly, posing substantial economic challenges to local pastoral livelihoods [12]. Current knowledge gaps persist regarding how native invasive plants affect local forage grass production and how to develop science-based management strategies.

The allelopathic capacity of native invasive plants simultaneously enhances their defensive adaptations and suppresses neighboring plants’ growth, serving as a key mechanism in invasion success [13]. Specifically, native invasive plants can produce allelochemicals, such as phytoalexins and alkaloids, which are rich in limonene and terpinene [14]. These substances have antibacterial properties and can alter the microbial assemblages, create favorable environmental conditions for the growth and establishment of invasive plants, and simultaneously inhibit the growth of surrounding plants [14]. Due to the negative impact of allelochemicals, these native invasive plants are also referred to as “toxic weeds” [15]. Currently, there is a large-scale expansion of invasive toxic weeds in the QXP region. The alpine meadows of the QXP represent a fragile ecosystem characterized by harsh bioclimatic conditions (short growing seasons, low temperatures, and high UV radiation) [16], unique biogeographical features with high endemism, and long-standing pastoral traditions [17]. In this region, many native plant have recently exhibited invasive behavior, particularly in degraded areas affected by changing grazing patterns and climate stressors [5]. The traditional nomadic grazing system, which previously maintained grassland equilibrium, has been disrupted by sedentarization and intensified land use, creating ecological niches where this native invasive species outcompetes key forage grasses [18]. For instance, the native invasive species, *Ligularia virgaurea* (Maxim.) Mattf., is a widely distributed “degradation indicator species” in this area [19,20]. Due to the presence of allelochemicals, it inhibits the germination and growth of forages in alpine meadows [20]. *Ligularia dictyoneura* (Franch.) Hand.-Mazz. has allelopathic effects, and its invasion and expansion have led to a significant reduction in vegetation density and the biomass of high-quality forages [18,21]. *Ligularia cymbulifera* (W. W. Sm.) Hand.-Mazz. is one of the main species in the southeastern QXP region. Studies have shown that the main allelochemicals of *L. cymbulifera* are p-hydroxybenzoic acid and its derivates [12]. These allelochemicals, as plant toxins, are released into the surrounding soil, jointly interfering with the germination and root elongation of neighboring plants to help *L. cymbulifera* gain an advantage in specific habitats. However, the specific impacts of various allelochemicals produced by native invasive plants on the germination and growth of different local forage seeds remain to be explored.

This study quantitatively assessed the inhibitory effects of three typical allelochemicals—p-hydroxybenzoic acid, benzoic acid, and caffeic acid—isolated from common native invasive plants [12,22,23], e.g., *L. cymbulifera*, *Ligularia sagitta* (Maxim.) Mattf. ex Rehder & Kobuski and *L. virgaurea*—upon four common forage grasses critical for restoring degraded alpine meadows on the QXP. The meadow community is dominated by *Kobresia* species, with common forage grasses including *Festuca elata* Keng ex E. B. Alexeev (FE), *Lolium perenne* L. (LP), *Medicago sativa* L. (MS), and *Trifolium repens* L. (TR) [24,25]. Four forage seeds were exposed to gradient concentrations of these compounds to assess species-specific responses in germination and seedling growth parameters. The selected species represent functionally distinct groups: Fabaceae (*M. sativa*, *T. repens*) enhance soil nitrogen fixation and organic matter content [26,27], while Poaceae (*F. elata*, *L. perenne*) provide rapid ground cover due to their high establishment capacity [28]. Germination assays elucidated allelochemical–forage growth relationships, enabling targeted species selection and sowing strategies for restoring degraded meadows on the QXP native plant invasion pressures. In summary, this study systematically examines the impacts of three typical allelochemicals from native invasive plants on germination and growth traits of four forages. By identifying allelochemical-tolerant species and analyzing their response differentials, we aim to provide a theoretical foundation for restoring alpine meadows facing unique ecological challenges. We propose the following key scientific questions: (1) Do different forage plants exhibit consistent responses to the same allelochemical? (2) Do different allelochemicals produce consistent effects on the same forage species? (3) Are the allelopathic effects dependent on both forage types and specific allelochemicals?

## 2. Results

### 2.1. Seed Germination Dynamics: Daily Germination Patterns Under Allelochemical Treatments

Different allelochemical concentrations distinctly influenced the daily germination patterns of the four forage species (Figure 1). Fabaceae seeds (*M. sativa*—MS and *T. repens*—TR) exhibited rapid germination, reaching peak germination (~20 seeds) by 3 days in controls (Figure 1c,d), while Poaceae seeds (*L. perenne*—LP and *F. elata*—FE) germinated slowly, completing germination after 2 weeks (Figure 1a,b). A low concentration of HA slightly promoted seed germination compared to the control. In contrast, medium-to-high CA and all BA treatments inhibited germination, with BA showing the strongest suppression.

### 2.2. Germination Parameters: Final Germination Rate (GR) and Germination Potential (GP) Across Concentrations

All allelochemicals significantly impacted germination rate (GR) and germination potential (GP) across four forage species. Under control conditions (water treatment), GR and GP ranked as follows: *T. repens* (TR) (86% GR, 83% GP) > *M. sativa* (MS) (78%, 77%) > *F. elata* (FE) (70%, 51%) > *L. perenne* (LP) (43%, 37%) (Figure 2). Allelochemicals exerted stronger inhibitory effects on GP than GR. Benzoic acid (BA) demonstrated the most potent inhibition: at BA500, it completely suppressed germination (GR = 0% for all species), while BA300 significantly reduced GR across all species (*p* < 0.05), with TR showing the strongest inhibition (GR = 0%). BA100 reduced TR’s GR by 49% versus control, minimally affected FE (−3%), and moderately impacted LP (−15%) and MS (−12%), though these changes were non-significant (*p* > 0.05) (Figure 2a). GP was severely inhibited, decreasing significantly at all BA concentrations (*p* < 0.05), while MS exhibited marginally greater resilience (16% reduction vs. control) (Figure 2b). Caffeic acid (CA) induced concentration-dependent GR suppression, with grasses (FE, LP) being more sensitive than legumes (TR, MS) (*p* < 0.05). Under 100–500 mg/L CA treatments, FE’s GR declined from 69% to 7% (control: 70%), and LP’s GR decreased from 48% to 28% (control: 43%). TR and MS showed only minor, non-significant reductions (*p* > 0.05) (Figure 2c). CA’s effect on FE’s GP mirrored GR trends, with significant inhibition at 300 and 500 mg/L (*p* < 0.05) (Figure 2d). P-Hydroxybenzoic acid (HA) significantly inhibited GR in TR (−44% vs. control) and FE (−55%) at high concentrations (*p* < 0.05), with negligible effects on other species (Figure 2e). HA300 and HA500 significantly suppressed GP in FE and TR (*p* < 0.05) (Figure 2f). Overall, BA exhibited the strongest inhibitory effects, CA preferentially inhibited grass germination over legumes, and HA’s effects showed no family-specific pattern.

### 2.3. Dose-Dependent Effects on Seedling Growth Traits: Root/Stem Length and Weight

The functional traits of seedlings (root length, root weight, stem length, stem weight) were significantly altered by the three allelochemicals (BA, CA, HA) in a concentration- and compound-dependent manner (Figure 3). Overall, BA exhibited the strongest inhibitory effect, followed by CA, while HA showed the weakest effect (seedlings remained viable at HA500, whereas *F. elata* seedlings failed to grow at CA500). Species-Specific Responses: *Festuca elata* & *Lolium perenne*: BA: Root length and root weight were significantly inhibited (*p* < 0.05) at all concentrations, except that stem traits were unaffected at BA100. CA: No significant effects at CA100; CA300 inhibited all traits except stem weight; CA500 inhibited all traits. HA: No significant effects at HA100/300; all traits inhibited at HA500 (*p* < 0.05). *Medicago sativa*: BA: BA100 promoted stem length and weight but inhibited root weight; BA300 inhibited root length, root weight, and stem weight; BA500 inhibited all traits (*p* < 0.05); CA: No significant effects at CA100 (*p* > 0.05); root length and root weight inhibited at CA300/500 (*p* < 0.05). HA: No significant effects at HA100/300 (*p* > 0.05); root length and root weight inhibited at HA500 (*p* < 0.05). *Trifolium repens*: BA: No response at BA100; all traits inhibited at BA300/500 (*p* < 0.05). CA: CA100 promoted root length; CA300 inhibited stem length; CA500 inhibited all traits (*p* < 0.05). HA: HA100 promoted root length; HA300 inhibited stem length; HA500 inhibited root length, root weight, and stem length (*p* < 0.05; stem weight unaffected).

### 2.4. Allelopathic Response Index (RI) of Seedling Functional Traits

The allelopathic response index was calculated based on the differences between each allelochemical treatment group and the control group (water treatment), enabling standardized comparisons of allelochemical effect strength across different experimental groups (e.g., varying plant species and allelochemical types). After treating four forage seeds with varying concentrations of three allelochemicals, the allelopathic response indexes (RIs) of functional traits were analyzed (Figure 4). Low concentrations (100 mg/L) slightly promoted seedling growth, as seen in the enhanced root growth and weight of *T. repens* and *M. sativa* under BA treatment (Figure 4a,d,g,j). However, increasing concentrations, particularly at 500 mg/L, intensified inhibitory effects (negative RI with increasing absolute values). The stem length and weight of *T. repens* and *M. sativa* remained largely unaffected by CA and HA treatments (Figure 4b,c,e,f,h,i,k,l). Medium and high concentrations of CA significantly inhibited the root and stem growth and weight accumulation of *L. perenne* (*p* < 0.05) (Figure 4b,e,h,k). Similarly, medium and high HA concentrations suppressed the root and stem growth and weight of *L. perenne* and *F. elata*, while low concentrations had minimal effects on these traits (*p* < 0.05) (Figure 4c,f,i,l). Overall, the results based standardized allelopathic response indexes also highlight concentration-dependent allelopathic responses, with low levels promoting growth and higher levels exerting stronger inhibitory effects.

### 2.5. Multi-Factors That Affect Germination and Growth Conditions

Finally, we employed a multifactorial mixed-effects model to comprehensively quantify the relative effects of different factors (plant species, allelochemical type, and concentration) at their respective levels (four plant species, three allelochemicals, and four concentration treatments) on plant germination and growth. Based on this integrated comparison, we were able to clearly identify the most influential allelochemicals and the most resistant forage species. We found that effect sizes varied considerably across the different treatments (Figure 5). Analysis of effect sizes revealed distinct response patterns of plant germination and growth to the three experimental factors (forage species, allelochemical type, and concentration). In general, germination conditions (germination rate and potential) were most affected in *M. sativa*, followed by *T. repens*, *L. perenne* and *F. elata* (MS > TR > LP > FE) (Figure 5a,b). Seedling growth traits (root length, root weight, stem length, stem weight) showed inconsistent responses, with the overall plant weight being greater in Fabaceae than in Poaceae; the length of roots and stems was greater in Poaceae than in Fabaceae (Figure 5c–f). Higher allelochemical concentrations consistently exerted stronger inhibitory effects, reflected in negative effect sizes that intensified with increasing concentration (*p* < 0.0001). Among the three allelochemicals, BA exhibited the strongest inhibitory impact, followed by CA and HA, demonstrating a consistent hierarchy in their phytotoxic effects (Figure 5c–f).

## 3. Discussion

### 3.1. Species- and Organ-Specific Responses to Allelochemicals

Our findings reveal that plants exhibit species-specific and organ-specific sensitivities when exposed to identical allelochemical stressors. *Trifolium repens* and *Medicago sativa*, both Fabaceae species, demonstrate markedly divergent responses—*Trifolium repens* shows high sensitivity to BA, while *Medicago sativa* maintains growth even at 300 mg/L BA. Under HA500 treatment, the two species from the Poaceae family, *F. elata* and *L. perenne*, showed different responses in stem growth. *Festuca elata* experienced significant reductions in both stem length and weight (*p* < 0.05), whereas the reductions observed in *L. perenne* were not statistically significant (*p* > 0.05). Such interspecific tolerance variations may stem from differential detoxification capacities (e.g., glutathione-S-transferase (GST) activity [29]) or stress-tolerance compound accumulation (e.g., ascorbic acid (AsA)). For instance, Guo et al. [30] discovered that under stress conditions, ascorbic acid (AsA) in *Medicago sativa* coordinates with multiple enzymes to convert reactive oxygen species (ROS) into non-toxic substances, thereby sustaining intracellular redox homeostasis. At the organ level, roots exhibit universally higher sensitivity than stems, identifying roots as primary targets of allelopathy. Li et al. [31] also found that the aqueous extract of *Artemisia argyi* H. Lév. & Vaniot significantly inhibited the growth of *Brassica rapa* (Lour.) Rupr, exhibiting greater inhibition on root length than on stem length. According to the Optimal Partitioning Theory (OPT), plants allocate biomass to organs acquiring the most limited resources, enhancing resource uptake to mitigate environmental stress [32]. Direct root exposure to allelochemicals triggers sacrificial root reduction to preserve photosynthetic stems and leaves. Additionally, aerial parts experience stress only after root uptake and translocation, amplifying root response intensity [33]. In summary, forage grass species exhibit differential sensitivity to the same allelochemical stress. Scientifically selecting resistant varieties constitutes a critical approach for grassland restoration in areas with serious native plant invasions. Particularly, root-level allelochemical resistance should be prioritized as a key breeding target for future forage cultivars.

### 3.2. Plant-Specific Sensitivity to Different Allelochemicals

We also found that different allelochemicals exert distinct effects on the same plant species. Allelopathic compounds can damage cell membrane integrity by enhancing peroxidase (POD) activity and modulating malondialdehyde (MDA) content—a lipid peroxidation marker [34]. However, the severity of such damage varies significantly among different allelochemicals. For example, BA typically exhibits stronger allelopathic effects than CA and HA, primarily due to its smaller molecular size (122.12 g/mol) and compact structure (single benzene ring + carboxyl group). This structural configuration enables BA to flexibly access enzyme active sites (e.g., POD), more readily disrupting key enzymatic functions [35]. Additionally, lipophilicity differences play a critical role: BA’s high log *p* value (1.87) facilitates penetration through lipid bilayers, promoting intracellular accumulation and interference with physiological processes (e.g., mitochondrial respiration, enzyme kinetics) [36]. In contrast, CA (featuring a benzene ring, two adjacent phenolic hydroxyl groups, and an acrylic acid side chain) and HA (with an added hydrophilic hydroxyl group) possess more complex structures that impede membrane permeation, thereby reducing their phytotoxicity [36]. This study provides experimental evidence confirming the differential phytotoxicity of various allelochemicals, suggesting that developing allelochemical-specific approaches could be an effective strategy against native plant invasions. For mono-dominant native invasive plants, their allelopathic effects may be primarily mediated by one major allelochemical (e.g., [23,37]), and identifying this key compound represents the first critical step for targeted control. However, it must be acknowledged that some invasive plants produce multiple allelochemicals (e.g., [12]), and co-occurrence of multiple invasive species can lead to complex soil allelochemical profiles. Significant knowledge gaps remain regarding the synergistic or antagonistic effects of multiple allelochemicals [38], and we hope that future research will elucidate their interaction mechanisms.

### 3.3. Concentration-Dependent Effects of Allelochemicals on Plants

Moreover, these allelochemicals BA, CA, and HA exhibit concentration-dependent inhibition of plant growth across concentrations of 100–500 mg/L, with allelopathic effects intensifying at higher concentrations but plant-growth promotion at lower concentrations. At higher concentrations, these compounds disrupt hormonal homeostasis in recipient plants, impairing growth-regulatory systems. Studies demonstrate that allelochemicals from rice leaves elevate indoleacetic acid oxidase (IAAO) activity in target weeds, reducing endogenous indole-3-acetic acid (IAA) levels and suppressing growth [39]. Phenolic allelochemicals (e.g., HA) interfere with IAA and gibberellin metabolism, inhibiting root and stem elongation in weeds [40]. At lower concentrations, *Festuca elata* showed slight root biomass increases under 100 mg/L CA and HA (*p* > 0.05), while *T. repens* exhibited significant root length and weight enhancement (*p* < 0.05) at 100 mg/L CA. This aligns with Ma et al.’s [41] findings on *L. virgaurea*’s biphasic allelopathy—stimulatory at low doses but inhibitory at high concentrations across forage species. These paradoxical responses are explained by hormesis—a toxicological phenomenon characterized by biphasic dose responses: low-concentration stimulation versus high-concentration toxicity [42]. At subtoxic levels (e.g., 100 mg/L), allelochemicals act as mild stressors that activate growth factor signaling pathways and cell survival genes, whereas elevated concentrations trigger cellular dysfunction and/or death [43]. This phenomenon provides an important insight: when assessing the risks of plant invasion, we must not be misled by short-term beneficial effects (e.g., low-concentration allelochemicals promoting plant biomass), as the accumulation of these compounds will ultimately lead to more significant negative impacts.

### 3.4. Practical Implications and Applications

Here we propose a management framework, beginning with the identification of dominant allelochemicals in invaded areas using allelochemical fingerprinting techniques—a comprehensive analytical approach that characterizes a plant’s unique profile of bioactive secondary metabolites responsible for its allelopathic interactions [44]. Then, when restoring degraded grasslands in areas threatened by native plant invasion, selecting forage plant varieties with broad hormesis windows (e.g., *M. sativa*) is recommended. Additionally, inherent growth differences exist between legume and grass seeds, indicating a restoration strategy based on functional complementary: grasses exhibit faster growth rates and greater expansion potential despite thinner root systems, while legumes play a critical role in nitrogen cycling through rhizobial symbiosis [45], serving as potential nitrogen reservoirs in restoration zones. Bi et al. [46] demonstrated that diversifide forage production systems—implemented with lower legume-grass seeding ratios combined with moderate nutrient management practices—constitute an effective strategy for achieving high yield and efficiency in cultivated grasslands. Based on these findings, we advocate implementing grass-legume polycultures in invaded areas (e.g., Xu et al. [47] demonstrated superior growth in 1:1 *L. perenne*-*T. repens* mixtures versus monocultures) to leverage complementary advantages—nitrogen fixation and rapid root establishment. For agricultural applications, the potent allelopathic effects of BA support its development as a bioherbicide, with concentrations as low as 100 mg/L producing significant inhibitory effects. However, managing biological invasions in field settings remains challenging due to complex antagonistic/synergistic interactions among multiple allelochemicals. Current research frontiers focus on microbial degraders (e.g., *Pseudomonas* spp. [48]) as biocontrol agents, given their ability to detoxify allelochemicals. While our study was laboratory-based, its theoretical findings hold significant potential for practical applications. However, there remains a substantial gap between theory and implementation, and we anticipate future applied research and field practices to further validate and expand upon our discoveries. Critically, our results demonstrate that aggressively expanding native invasive plants pose invasive threats comparable to exotic species due to their strong allelopathy. Thus, we propose classifying such native super-dominant plants as invasive species within management frameworks to improve ecosystem governance.

## 4. Materials and Methods

### 4.1. Experimental Plants and Reagents

The study examined four representative forage species—*M. sativa* “WL440HQ” (Fabaceae), *T. repens* “Haifa” (Fabaceae), *L. perenne* “Maidi” (Poaceae), and *F. elata* “Airui 3” (Poaceae)—all sourced from Beijing Zhengdao Technology Co., Ltd (Beijing, China). The selected forage species are dominant in heavily grazed alpine meadows where native plant invasions most severely impact pastoral production [17,20]. As these cultivated forage species are widely adopted for regional pasture rehabilitation due to their adaptability and yield, our study targets their responses to allelochemicals to directly inform economic grassland restoration strategies. While laboratory conditions simplify edaphoclimatic variables, this approach allows standardized assessment of allelopathic effects on management-relevant species, avoiding ecological risks associated with introducing non-natives into natural meadows [12,49]. The use of standardized commercial cultivars guarantees genetic homogeneity across experimental samples, effectively controlling for confounding effects of genetic variation on allelopathic response assessments [12].

The study selected three allelochemicals as experimental reagents: BA: benzoic acid, CA: caffeic acid, and HA: p-hydroxybenzoic acid (all analytical grade). These compounds were chosen because they have been identified as main allelochemicals in common native plants, primarily from *Ligularia* species, within the Qinghai-Xizang Plateau region (e.g., [12,22,23]). They also represent a major class of allelochemicals characterized by small molecular size, structural stability, and persistence in soil environments where they can accumulate to phytotoxic levels.

### 4.2. Experimental Designs

The experimental procedure consisted of two main phases: solution preparation and germination testing. Experimental concentrations (100–500 mg/L) were determined through pre-tests and validated against in situ measurements of rhizosphere soil concentrations (105 mg/kg–434 mg/kg) in native invasive plants, ensuring ecological relevance while maintaining detectable phytotoxicity. For solution preparation, 1 g of HA was dissolved in 4 mL of 75% ethanol with vigorous stirring until complete dissolution. This solution was then transferred to 1000 mL of pure water to prepare a 1 g/L aqueous stock solution, which was subsequently diluted to final concentrations of 500, 300, and 100 mg/L (denoted as HA500, HA300, HA100). Identical preparation methods were applied to BA and CA, yielding solutions labeled BA500-BA100 and CA500-CA100, respectively, all stored at 4 °C.

Based on long-term local meteorological records and our field measurements, we simulated the diurnal temperature variations at 3000 m a.s.l. [10], which typically range between 15 and 26 °C during the growing season (July–August) in alpine meadows (elevation: 3000–3500 m). Germination experiments were conducted in controlled climate chambers using the Petri dish filter paper method under standardized conditions: humidity is 60%, 25 °C during daytime/20 °C during nighttime with 12 h photoperiods at 10,000 lx light intensity [12,49]. Seeds underwent surface sterilization (75% ethanol for 5 min followed by triple rinsing with distilled water) before selection of uniform, plump specimens [49]. The experimental setup involved placing 25 seeds on double-layered 9 mm filter paper in each Petri dish, with daily applications of 5 mL initial treatment solution followed by 1–2 mL maintenance doses. To eliminate the effects of fixed positions, Petri dishes were randomly repositioned during each watering event, and damaged seeds were removed using flame-sterilized forceps to prevent microbial contamination. Concurrently, dishes were arranged singly without overlap within the incubator to ensure uniform and sufficient light exposure. The factorial design included 4 plants × 3 allelochemicals × 4 concentrations (with distilled water controls: CK, 0 mg/L) × 4 replicates. Each forage species was subjected to treatments with four concentrations of three allelochemicals, with four biological replicates per treatment combination. The experiment was terminated when no new germination occurred for two consecutive days, a criterion consistently met within the 14-day monitoring period. Germinated seeds were counted daily at fixed intervals. For *M. sativa* and *T. repens*, the germination rate was calculated at day 14, while germination potential was determined at day 3, corresponding to their peak germination phase observed experimentally. Due to delayed germination kinetics in Poaceae, *F. elata* and *L. perenne* had germination rate assessed at day 14 and germination potential at day 7. The germination monitoring period extended from the initial emergence of seed germination until the complete formation of the first true leaves. Post-germination measurements included root/stem lengths (measured with ruler) and fresh weights (using 0.0001 g precision balance), with three seedlings randomly selected per replicate (12 seedlings total per treatment) for subsequent averaging.

### 4.3. Statistical Analyses

For seed germination analysis, we calculated germination rate (GR) and germination potential (GP) using the following formulas:
(1)GR=(M1/M)×100%
(2)GP=(M2/M)×100%
where M1 is the number of normally germinated seeds at the final count, M2 is the number of seeds germinated during peak germination, and M is the total number of seeds.

The allelopathic effect of allelochemical on seedling growth traits was quantified using the allelopathic response index (RI) as described by Williamson et al. [50] as the standardized comparison parameters among different plants.



(3)
RI=1−C/T(T≧C)orRI=T/C−1(T<C)



Here, C represents the control value (distilled water treatment), and T represents the treatment value (allelochemical exposure) [51]. An RI < 0 indicates inhibition, while RI > 0 suggests promotion, with the absolute RI value reflecting the allelopathic intensity [52].

All statistical analyses were performed in R v.4.3.3 [53]. We visualized daily germination dynamics of the four forage species under three allelochemical treatments using cumulative germination curves generated with the ggplot2 [54] and ggsci [55] packages. Differences in GR, GP, and RI of growth traits were assessed via boxplot analyses, while multivariate effects of allelochemical treatments on plant traits were analyzed using heatmap analyses (pheatmap package [56]). Due to violation of the normality and homoscedasticity assumption in data distribution (*p* < 0.01), we adopted a more conservative non-parametric test (Mann–Whitney U test) for pairwise comparisons. Linear mixed-effects models (lme4 package) evaluated the relationships among plant species (*M. sativa*, *T. repens*, *L. perenne*, and *F. elata*), allelochemical concentration (0, 100, 300 and 500 mg/L), and allelochemical type (BA, CA and HA) in relation to seed germination and seedling growth (dependent variables: germination rate, germination potential, root length, root weight, stem length, stem weight), respectively. Experimental samples were treated as a random effect to account for biological variation. Based on these multivariate models, we can investigate the combined effects of multiple factors on plant germination and growth.

## 5. Conclusions

Our study establishes that three dominant allelochemicals (benzoic acid, caffeic acid, p-hydroxybenzoic acid) from typical native invasive plants on the Qinghai-Xizang Plateau exhibit concentration-dependent hormetic effects on four key forage species, with low concentrations stimulating germination but higher concentrations causing progressive inhibition. The results reveal distinct species-specific vulnerability patterns: *F. elata* (FE) showed the greatest germination sensitivity (FE > LP > TR > MS), while *T. repens* (TR) demonstrated consistently high sensitivity across all seedling growth parameters (root length, biomass, and shoot metrics). Notably, root systems exhibited significantly greater sensitivity than stems (*p* < 0.01), highlighting organ-specific responses to allelopathic stresses. These findings collectively demonstrate that native invasive plants generate substantial allelopathic pressure on local economic pastures through multiple biochemical pathways. Most importantly, by revealing that native invaders can exert ecological impacts comparable to exotic invasive species, our results challenge conventional invasion biology paradigms and underscore the urgent need to develop targeted management strategies for native biological invasions in vulnerable alpine ecosystems. Our study provides valuable theoretical guidance for local pasture restoration and management, demonstrating that optimized forage selection strategies offer a sustainable solution for addressing native plant invasions.

## Figures and Tables

**Figure 1 plants-14-03506-f001:**
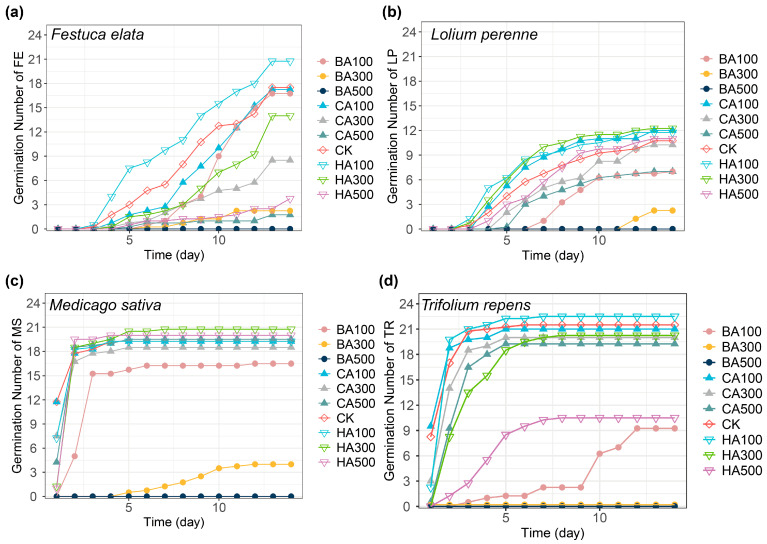
Effect of different concentrations of the three allelochemicals on the number of plant seeds germinated per day. (**a**): FE: *Festuca elata* Keng ex E. B. Alexeev; (**b**): LP: *Lolium perenne* L.; (**c**): MS: *Medicago sativa* L.; (**d**): TR: *Trifolium repens* L. (BA300 and BA500 induced complete germination suppression in TR (0% GR); positional offset applied for visual distinction). The letters on the right side of each figure represent the following: BA: benzoic acid; CA: caffeic acid; HA: p-hydroxybenzoic acid; CK: pure water. Numbers represent different concentrations; numbers 100: 100 mg/L; numbers 300: 300 mg/L; and numbers 500: 500 mg/L.

**Figure 2 plants-14-03506-f002:**
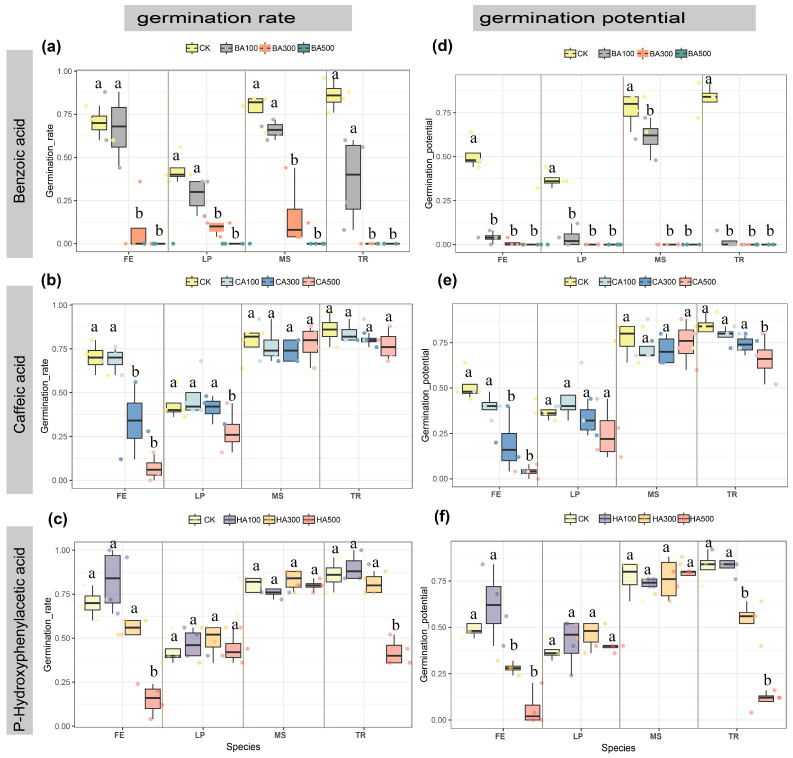
Effects of each concentration of the three phenolic acids on seed germination rate and germination potential. (**a**) Effects of benzoic acid concentration on germination rate. (**b**) Effects of caffeic acid concentration on germination rate. (**c**) Effects of p-hydroxybenzoic acid concentration on germination rate. (**d**) Effects of benzoic acid concentration on germination potential. (**e**) Effects of caffeic acid concentration on germination potential. (**f**) Effects of p-hydroxybenzoic acid concentration on germination potential. The letters at the top and on the right side of each figure represent the following: BA: benzoic acid; CA: caffeic acid; HA: p-hydroxybenzoic acid; CK: pure water; FE: *Festuca elata*; LP: *Lolium perenne*; MS: *Medicago sativa*; TR: *Trifolium repens*. Numbers represent different concentrations; numbers 100: 100 mg/L; numbers 300: 300 mg/L; and numbers 500: 500 mg/L. Data presented as median ± IQR (*n* = 4). Different superscript letters indicate statistically significant differences (*p* < 0.05) compared to the control group (CK).

**Figure 3 plants-14-03506-f003:**
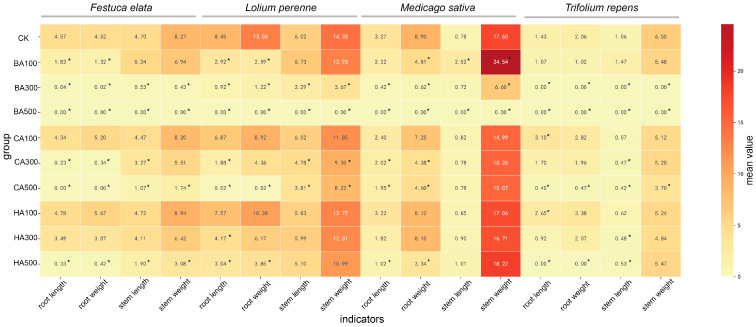
Dose-dependent effects of allelochemicals on seedling functional traits. Heatmap displays mean values of root/stem length and weight under BA, CA, and HA treatments (100–500 mg/L). Color intensity scales with trait magnitude (darker = higher). * indicate significant differences versus CK control (*p* < 0.05).

**Figure 4 plants-14-03506-f004:**
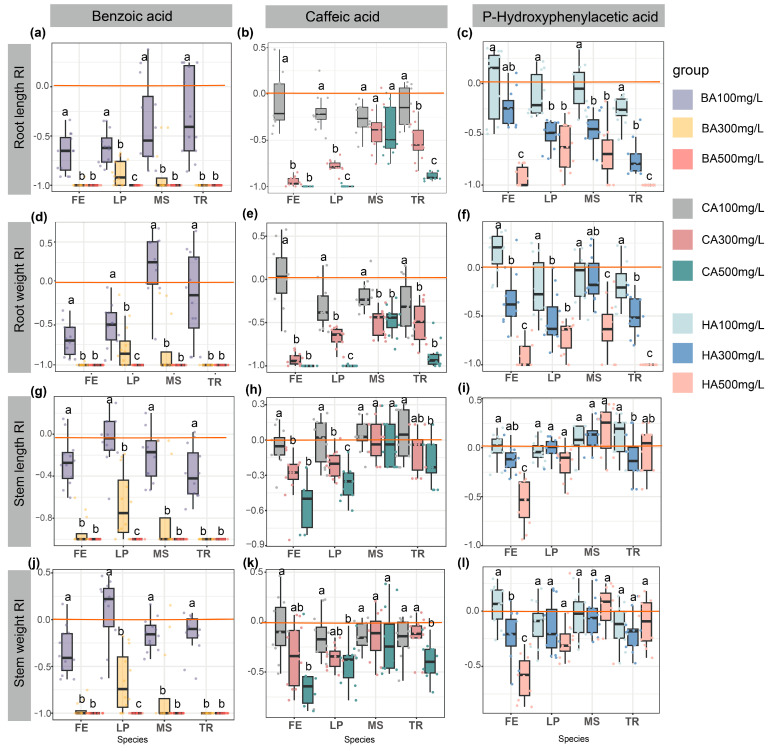
Dose-dependent effects of phenolic acids on functional traits of four forage species. Box plots illustrate the allelopathic response index (RI) of (**a**–**c**) root length, (**d**–**f**) root weight, (**g**–**i**) stem length, and (**j**–**l**) stem weight in *Festuca elata* (FE), *Lolium perenne* (LP), *Medicago sativa* (MS), and *Trifolium repens* (TR) under treatments with: Left column: Benzoic acid (BA) at 100, 300, 500 mg/L; Middle column: Caffeic acid (CA) at 100, 300, 500 mg/L; Right column: p-Hydroxyphenylacetic acid (HA) at 100, 300, 500 mg/L. Data presentation: Boxes show median ± IQR (n = 4 replicates, 25 seeds each). Negative RI values indicate trait suppression (e.g., BA 500 mg/L induced complete root inhibition in TR, RI = −1), while positive values signify stimulation. Different superscript letters indicate statistically significant differences (*p* < 0.05). The red horizontal line in the figure indicates the baseline of RI = 0.

**Figure 5 plants-14-03506-f005:**
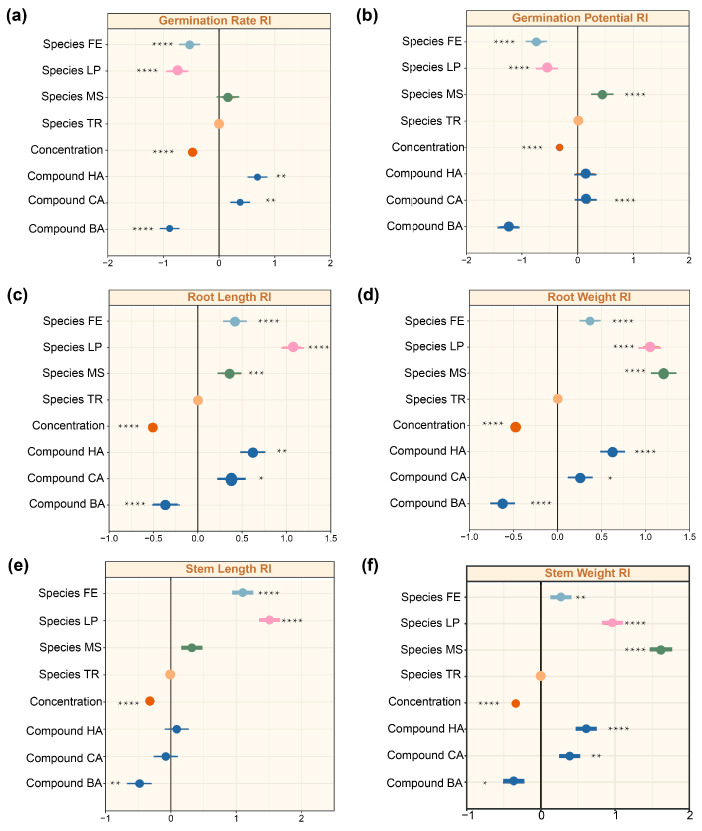
Effects of forage grass species, compound types, and concentrations on plant traits based on linear mixed effects model (LMMS). (**a**) Germination rate as the response variable. (**b**) Germination potential as the response variable. (**c**) Root length RI as the response variable. (**d**) Root weight RI as the response variable. (**e**) Stem length RI as the response variable. (**f**) Stem weight RI as the response variable. The letters on each figure represent the following: RI: the allelopathic response index, BA: benzoic acid, CA: caffeic acid, HA: p-hydroxybenzoic acid, CK: pure water. TR: *Trifolium repens*, MS: *Medicago sativa*, FE: *Festuca elata*, LP: *Lolium perenne*. Correlation significance: * *p* < 0.05; ** *p* < 0.01; *** *p* < 0.001; **** *p* < 0.0001. The factor ‘concentration’ in the model integrates the four tested levels (0, 100, 300, and 500 mg/L). The compound concentrations are now labeled in orange, the compound types are indicated in dark blue, and the four different forage species are distinguished using four distinct colors. The meaning of the black line, representing the RI = 0 baseline.

## Data Availability

Data are contained within the article.

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
