# Peer review of "Allelopathic Effects of Dominant Native Invaders on Forage Establishment: Implications for Alpine Meadow Restoration on the Qinghai-Xizang Plateau"

_plants, 2025, doi:10.3390/plants14223506_

Round 1

Reviewer 1 Report (Previous Reviewer 2)

Comments and Suggestions for Authors

Please keep the format of the subheadings consistent with the journal guidelines, as some words currently begin with lowercase letters while others begin with uppercase letters.

Comments on the Quality of English Language

 The English could be improved to more clearly express the research.

Author Response

Comments 1: [ Please keep the format of the subheadings consistent with the journal guidelines, as some words currently begin with lowercase letters while others begin with uppercase letters. ]

Response 1:

[We thank the reviewer for this comment. We have carefully checked all subheadings and reformatted them to ensure full consistency with the journal's guidelines regarding capitalization. ]

Comments 2: [The English could be improved to more clearly express the research.]

Response 1: [We thank the reviewer for this valuable comment. We agree that improving the clarity and quality of the English language is crucial for effectively communicating our research.

In response, we have thoroughly revised the entire manuscript to enhance the language, grammar, and overall readability. We have paid special attention to simplifying complex sentences, clarifying ambiguous expressions, and ensuring the logical flow of ideas is clear and precise.

For instance, in the Methodology section (Lines 131-132), we have rephrased the description of our experimental procedure from “[Low-to-medium concentrations of HA and low-concentration CA slightly promoted germination, with daily counts exceeding controls.]” to “[A low concentration of HA slightly promoted seed germination compared to the control.]” to make it more concise and understandable. Similar revisions have been made throughout the paper.

We believe these comprehensive edits have significantly improved the manuscript's clarity, and we are grateful for the reviewer's suggestion.]

Reviewer 2 Report (New Reviewer)

Comments and Suggestions for Authors

The MS is well written and structured. I have enjoyed reading it. However, there are minors to be done prior to acceptance:

L73-74. change microbial ecosystem to microbial assemblages

L82. native (invasive) species, pls. delete invasive there. It is clear and You have again invasive couple of words afterwards

Latin names, once introduced in MS can be used as short (e.g. L. cymbulifera), if this are not in graphs labels or at the beginning of the sentences. Also, where firstly introduced use full name and authorities (e.g. L102. Ligularia saggita missing authorities). Also, put all latin names in italic also in references.

L114. remove amid

L131-132. Figures are mismatched in text. 1a,b should be 1c, d and vice versa, or replace the position in graph.

In Figure label, remove each and apply different concentrations

In some figures it is specie should be species

L426. in situ should be in italic

English is partly odd. (e.g. L257 Analysis of effect sizes revealed). Please, make it rather easy and clearly to understand what it is referred to.

Author Response

The MS is well written and structured. I have enjoyed reading it. However, there are minors to be done prior to acceptance:

Response:

We sincerely thank the reviewer for their positive assessment of our manuscript and for their valuable time in providing constructive feedback. We have carefully addressed all the minor points raised in the comments below and have incorporated the corresponding revisions throughout the manuscript, with all changes highlighted in yellow in the resubmitted version. Our detailed point-by-point responses are provided below.

Comments 1: [L73-74. change microbial ecosystem to microbial assemblages]

Response1: We thank the reviewer for the suggestion. We have changed "microbial ecosystem" to "microbial assemblages" in lines 73-74 of the revised manuscript.

Comments 2: [L82. native (invasive) species, pls. delete invasive there. It is clear and You have again invasive couple of words afterwards]

Response 2: We thank the reviewer for this correction. We have deleted "invasive" from the phrase "native (invasive) species" in line 82 of the revised manuscript.

Comments 3: [ Latin names, once introduced in MS can be used as short (e.g. L. cymbulifera), if this are not in graphs labels or at the beginning of the sentences. Also, where firstly introduced use full name and authorities (e.g. L102. Ligularia saggita missing authorities). Also, put all latin names in italic also in references.]

Response 3: We thank the reviewer for these important reminders on taxonomic nomenclature. We have carefully revised the manuscript to ensure that:

  • All Latin names are italicized throughout the text and references.
  • Full names with authorities are provided at first mention (e.g., Ligulariasagitta (Maxim.) Mattf. ex Rehder & Kobuski has been corrected on L102).
  • Abbreviated names (e.g., L. sagitta) are used subsequently where appropriate, except in figure labels or at the beginning of sentences.

Comments 4: [L114. remove amid]

Response 4: We thank the reviewer for the suggestion. We have removed "amid" from the manuscript as requested.

Comments 5: [L131-132. Figures are mismatched in text. 1a,b should be 1c, d and vice versa, or replace the position in graph.]

Response 5: We thank the reviewer for pointing out this discrepancy. We have corrected the figure citations in the text (lines 131-132) to accurately match the panels in Figure 1.

Comments 6: [In Figure label, remove each and apply different concentrations]

Response 6: Thank you for the suggestion. We have replaced "each" with "different concentrations" in the figure labels as recommended.

Comments 7: [In some figures it is specie should be species]

Response 7: We thank the reviewer for catching this error. We have corrected "specie" to "species" in all figures throughout the manuscript.

Comments 8: [L426. in situ should be in italic]

Response 8: Thank you for this correction. We have italicized "in situ" on line 422 of the revised manuscript.

Comments 9: [English is partly odd. (e.g. L257 Analysis of effect sizes revealed). Please, make it rather easy and clearly to understand what it is referred to.]

Response 9: We thank the reviewer for this suggestion. We have rephrased the sentence to improve clarity and readability. The revised text now reads:
"The analysis of effect sizes showed distinct response patterns of plant germination and growth to the three experimental factors (forage species, allelochemical type, and concentration)." Additionally, we have made corresponding improvements to similar expressions throughout the manuscript to ensure consistency in terminology and clarity.

Reviewer 3 Report (New Reviewer)

Comments and Suggestions for Authors

The manuscript provides detailed and valuable data on allelochemical effects of substances derived from invasive native species on germination and growth of some Fabaceae and Poaceae species. Three types of allelochemicals of different concentrations (benzoic acid, caffeic acid and p-hydroxybenzoic acid) were tested. Low concentrations of allelochemicals stimulated germination, and higher concentrations caused progressive inhibition. The benzoic acid had the strongest allelopathic effect. The roots growth was more inhibited than stems.

There are some gaps that you need to fill in, especially in terms of methodology and results. Some results need to be reorganized for a better understanding. To further strengthen your paper, I suggest the following:

  1. Firstly, I would like to know if it is the final version for peer review. It seems that there are some comments made on the text that are not visible, and in some places the text is marked red.
  2. Rewrite the references in order to respect the authors guidelines for this journal
  3. Mention all the Ligularia species that you use for allelochemicals
  4. Some data are misspelled (lines 131-134, 139, etc) - see comments in the text
  5. Use different color to represent BA 300 or BA500, as it is hard to differentiate between them (See graphs – Figure 2)
  6. Explain the dose-dependent effects of CA and HA on Medicago sativa
  7. Delete lines 241-248. Are similar to those above (see comments in the text)
  8. Redraw the graphs in Figure 5 to make them easier to understand (see comments in the text)
  9. The text in lines 288-290 is incomprehensible. Rewrite it.
  10. Use SI units.
  11. How many petri dishes did you used? one per each species/each treatment?
  12. The formulas for germination rate and germination potential are misspelled.
  13. Mention all the abbreviations used in the text

Author Response

Comments: [The manuscript provides detailed and valuable data on allelochemical effects of substances derived from invasive native species on germination and growth of some Fabaceae and Poaceae species. Three types of allelochemicals of different concentrations (benzoic acid, caffeic acid and p-hydroxybenzoic acid) were tested. Low concentrations of allelochemicals stimulated germination, and higher concentrations caused progressive inhibition. The benzoic acid had the strongest allelopathic effect. The roots growth was more inhibited than stems.

There are some gaps that you need to fill in, especially in terms of methodology and results. Some results need to be reorganized for a better understanding. To further strengthen your paper, I suggest the following:]

Response: [We sincerely thank the reviewer for the positive evaluation of our work and for providing these constructive suggestions to help us improve the manuscript. We have carefully addressed all the points raised, with particular attention to enhancing the methodology and results sections. Detailed revisions have been made to improve clarity and organization, with all revisions highlighted in yellow in the resubmitted manuscript. Our point-by-point responses are provided below.

In addition, we have also provided detailed replies to the specific comments and annotations included in the PDF file provided by the reviewer. These have been addressed both in the revised manuscript and in the response letter accordingly.]

Comments 1: [Firstly, I would like to know if it is the final version for peer review. It seems that there are some comments made on the text that are not visible, and in some places the text is marked red.]

Response 1: We thank the reviewer for pointing this out. The red highlighted text in the manuscript represents revisions made in response to the previous round of reviewer comments. This version has been fully revised to address all previous feedback and is now resubmitted for your consideration. We confirm that all tracked changes from the previous revision cycle have been accepted and incorporated into the current document.

Comments 2: [Rewrite the references in order to respect the authors guidelines for this journal]

Response 2: We thank the reviewer for this comment. We have now reformatted the entire reference list to ensure full compliance with the journal's author guidelines. All references have been verified and updated to meet the required citation style.

Comments 3: [Mention all the Ligularia species that you use for allelochemicals]

Response 3: We thank the reviewer for this comment. The Ligularia species used for allelochemical testing in this study are: Ligularia sagitta (Maxim.) Mattf. ex Rehder & Kobuski, Ligularia virgaurea (Maxim.) Mattf., Ligularia cymbulifera (W.W.Sm.) R.Mathieu.

These have been clearly specified in the [Methods/Results] section of the revised manuscript (lines 102).

Comments 4: [Some data are misspelled (lines 131-134, 139, etc) - see comments in the text]

Response 4: We thank the reviewer for identifying these inconsistencies. The mismatched order between the figure panels and the corresponding text descriptions (previously in lines 131-134 and 139) has now been corrected in the revised manuscript. The text and figures have been aligned to ensure accurate referencing.

Comments 5: [Use different color to represent BA 300 or BA500, as it is hard to differentiate between them (See graphs – Figure 2)]

Response 5: We thank the reviewer for this suggestion. We have revised the color scheme in Figure 2 to use more distinct colors for BA300 and BA500, ensuring they are easily distinguishable in the graphs.

Comments 6: [Explain the dose-dependent effects of CA and HA on Medicago sativa]

Response 6: We thank the reviewer for this suggestion. The dose-dependent effects of CA and HA on Medicago sativa have been explained in the revised text (lines 197-199). The specific details are as follows:

[CA: No significant effects at CA100 (P > 0.05); root length and root weight inhibited at CA300/500 (P < 0.05). HA: No significant effects at HA100/300 (P > 0.05); root length and root weight inhibited at HA500 (P < 0.05).]

Comments 7: [Delete lines 241-248. Are similar to those above (see comments in the text)]

Response 7: We thank the reviewer for this observation. The redundant content in lines 241-248 has been deleted from the revised manuscript as suggested.

Comments 8: [Redraw the graphs in Figure 5 to make them easier to understand (see comments in the text)]

Response : 8We thank the reviewer for these detailed suggestions. We have carefully revised Figure 5 by:
(1) applying distinct colors to differentiate the four forage species,
(2) correcting all misspelled labels, and
(3) completing any incomplete textual elements.
The redesigned graphs now provide clearer visual representation of the results.

Comments 9: [The text in lines 288-290 is incomprehensible. Rewrite it.]

Response 9: Thank you for your suggestion. We have rephrased the indicated sentence for clarity. The revised version has been incorporated into the manuscript at the same location. The specific details are as follows:

[We found that effect sizes varied considerably across the different treatments (Figure 5). Analysis of effect sizes revealed distinct response patterns of plant germination and growth to the three experimental factors (forage species, allelochemical type, and concentration). (see lines 248-251)]

Comments 10: [Use SI units.]

Response 10: We thank the reviewer for this important reminder. We have thoroughly checked the entire manuscript and replaced all relevant units with the International System of Units (SI).

Comments 11: [How many petri dishes did you used? one per each species/each treatment?]

Response 11: We appreciate the opportunity to clarify this. Our experimental design was fully crossed, utilizing 4 seed species, 3 allelochemicals, and 4 concentration levels with 4 biological replicates for each combination. Consequently, the total number of petri dishes used was 4 × 3 × 4 × 4 = 192.

Comments 12: [The formulas for germination rate and germination potential are misspelled.]

Response 12: We thank the reviewer for correcting this error. We have replaced the formulas in the manuscript with the correct ones as provided. (see lines 460-461)

GR =(M1/M)×100%

GP =(M2/M)×100%

Comments 13: [Mention all the abbreviations used in the text]

Response 13: Thank you for this suggestion. We have now added a list of all abbreviations used in the manuscript. The complete list includes: [BA - benzoic acid, CA - cinnamic acid, HA - hydroxyacetic acid, FE - Festuca elata Keng ex E. B. Alexeev., LP - Lolium perenne L., MS - Medicago sativa L., TR - Trifolium repens L., GR - germination rate, GP - germination potential, RI - Response Index, LMMS - linear mixed effects model, QXP - Qinghai-Xizang Plateau].

Comments 14: [revise all the references according to the authorguidelines ]

Response 14: We thank the reviewer for this reminder. We have carefully checked and revised the format of all references in the manuscript to ensure they fully comply with the journal's author guidelines.

Comments 15: [without capital letters]

Response 15: thank you for this comment. we have now revised the text as suggested. (see lines 37-38)

This manuscript is a resubmission of an earlier submission. The following is a list of the peer review reports and author responses from that submission.

Round 1

Reviewer 1 Report

Comments and Suggestions for Authors

The manuscript in general is interesting and addresses a novel field with a potential application of its field. However, the manuscript is far from being publishable in Plants in its current form. 
General comments on each section and specific comments are attached to the manuscript itself.

Good luck to the authors!

Comments on the Quality of English Language

The manuscript generally employs appropriate scientific terminology and maintains a professional tone. However, several issues regarding grammatical accuracy and syntactic clarity should be addressed. Specifically, the Discussion section contains instances of subject–verb disagreement (e.g., “this responses aligns” should read “these responses align”), as well as unnecessarily long and complex sentences that may hinder reader comprehension. Additionally, certain key observations—such as the stimulatory effects of low concentrations of allelochemicals—are not sufficiently discussed, which weakens the overall interpretative depth of the work.

It is strongly recommended that the manuscript undergo careful revision by a native English speaker or a professional language editing service to ensure grammatical correctness, improved readability, and stylistic consistency across sections. This would enhance the manuscript’s clarity and accessibility for the international scientific audience.

Reviewer 2 Report

Comments and Suggestions for Authors

Abstract

The study addresses a relevant issue but lacks context on the scale of invasion.

The choice of allelochemicals and forage species needs clearer justification.

Results would be improved from quantitative details.

Differences in species’ responses are noted without explaining the mechanisms.

The restoration framework is promising, but feasibility in real conditions is unclear.

More evidence is needed to support the functional complementarity concept.

Overall, concise but could improve by adding practical and mechanistic insights.

Introduction

The introduction could better emphasize the specific gap regarding native invasive plants.

The economic and ecological impacts are well presented, though more recent or regional data might strengthen relevance.

The explanation of allelopathy and allelochemicals is somewhat dense; simplifying could improve clarity.

The link between allelochemicals and invasive success could be more emphasis on how this relates specifically to native invaders would help.

The background on alpine meadows and native toxic weeds needs clearer connections between allelochemicals and forage degradation could be emphasized.

The research gap is well defined, though the rationale for selecting the three allelochemicals and four forage species could be more explicit.

The study questions are clear and focused, providing a solid foundation for the research objectives.

Overall, the introduction could be more concise and focused on the native invasion and the allelopathy link.

Results

The results section lacks a clear and logical flow, making it challenging to follow the progression of findings. Transitions between different experiments and traits are abrupt and confusing.

Numerical results such as percentage reductions and germination rates are presented without sufficient interpretation of their biological or ecological significance, which weakens the impact of the findings.

Although figures are referenced, the text often does not adequately explain what each panel shows or how to interpret the data, and some figure captions are unclear or incomplete.

The use of species names and abbreviations is inconsistent, leading to potential confusion for readers.

Statistical results lack detailed reporting of p-values, confidence intervals, or test statistics, which are necessary for evaluating the robustness of the conclusions.

Percentages are frequently given without proper baseline values or comparative context, limiting reader understanding of the actual effects.

The differences between the allelochemicals’ effects are noted but not systematically compared or clearly summarized, missing an opportunity to highlight key contrasts.

The explanation of the allelopathic response index (RI) is vague, and its ecological implications are not fully discussed, leaving readers uncertain about its significance.

The section discussing effect sizes and mixed-effects models is underdeveloped and does not provide enough detail to clarify how these analyses support the results.

There are several language and grammar errors throughout the section that reduce readability and detract from the professionalism of the manuscript.

Discussion

The discussion is heavily descriptive but lacks critical analysis of limitations or alternative explanations for the observed patterns.

Many claims rely on previous studies without sufficiently integrating how the current results advance or challenge existing knowledge.

The proposed restoration strategies remain speculative without field validation data included in the study.

Some key terms (e.g., “functional complementarity,” “allelochemical fingerprinting”) are introduced without clear operational definitions or methodological details.

There is occasional repetition of ideas, which could be streamlined for clarity and conciseness.

The discussion assumes causality in some relationships (e.g., seed coat thickness as the main factor for Fabaceae tolerance) without direct experimental evidence in this study.

The reliance on pH differences to explain BA’s effects is simplistic and should consider other chemical or physiological factors.

Suggestions for future research are broad and lack specific, actionable steps that build directly on the present findings.

The section would benefit from a more explicit linkage between the observed organ-specific responses and practical implications for restoration.

Some references cited to support key points appear outdated or tangential and should be reassessed for relevance.

Materials and Methods

The selection of forage species is well justified ecologically, but no information is given on seed provenance or genetic variation that could affect results.

The choice of phenolic acids is appropriate, yet the rationale for excluding other relevant allelochemicals is not discussed.

Preparation of stock solutions includes ethanol as a solvent, but potential solvent effects on seeds are not addressed or controlled for.

The germination test conditions are standard but may not fully replicate field environments, limiting ecological relevance.

The sample size (25 seeds per replicate) is reasonable, but replication details could be clearer regarding randomization and blocking.

The use of the allelopathic response index (RI) is appropriate, though its limitations and assumptions are not acknowledged.

Statistical methods are generally sound, but the justification for using non-parametric tests and mixed models lacks detail on model diagnostics or assumptions.

No mention of controlling or measuring environmental variables like humidity or light uniformity in growth chambers, which could influence results.

The use of cumulative germination curves and radar plots could benefit from a clearer explanation of data interpretation.

Potential interaction effects between different allelochemicals are not explored in the experimental design.

Conclusion

Please add a clear and concise conclusion under a separate heading summarizing the main findings and their implications to enhance the overall clarity and impact of the paper. 

Comments on the Quality of English Language

The English could be improved to more clearly express the research.
